# The Rare Case of a COVID-19 Pregnant Patient with Quadruplets and Postpartum Severe Pneumonia. Case Report and Review of the Literature

**DOI:** 10.3390/medicina57111186

**Published:** 2021-11-01

**Authors:** Răzvan Socolov, Mona Akad, Maricica Păvăleanu, Diana Popovici, Mădălina Ciuhodaru, Roxana Covali, Fawzy Akad, Ioana Păvăleanu

**Affiliations:** 1Department of Obstetrics and Gynecology, Elena Doamna Obstetrics and Gynecology University Hospital, 700398 Iasi, Romania; socolov.razvan@gmail.com (R.S.); maricicapavaleanu@yahoo.com (M.P.); dianapopovici1964@yahoo.com (D.P.); mciuhodaru@yahoo.com (M.C.); rcovali@yahoo.com (R.C.); ioana_pavaleanu@yahoo.com (I.P.); 2Department of Obstetrics and Gynecology, Grigore T. Popa University of Medicine and Pharmacy, 700115 Iasi, Romania; akad.Fawzy@yahoo.com

**Keywords:** multiple pregnancy, COVID-19, postpartum pneumonia

## Abstract

*Background and Objectives*: The multiple pregnancies associated with COVID-19 is a new and difficult condition to manage. The prognosis for rapid deterioration after the cesarean delivery is difficult to assess and needs close interdisciplinary follow-up due to pregnancy and postpartum-related changes. *Materials and Methods*: We report the case of a 37-year-old primigesta primipara patient who was admitted to “Elena Doamna” Clinical Hospital of Obstetrics and Gynecology at 33 weeks and 3 days of gestation with high-grade multiple pregnancies (triplets) for threatened premature birth associated with COVID-19. The patient had a history of surgically corrected atrial septal defect during childhood and currently is known to have paroxysmal supraventricular tachycardia. Tocolysis was ineffective and the decision to perform a cesarean operation was made. The diagnosis was established: primigesta, primipara, at 34 weeks of gestation, high-grade multiple pregnancy with triplets, intact membranes, threatened premature birth, surgically corrected atrial septal defect, paroxysmal supraventricular tachycardia, infection with COVID-19. The patient underwent a cesarean intervention and treatment for COVID-19 pneumonia. The intervention took place at 33 weeks and 4 days of gestation resulting in four newborns with weights between 1400 g and 1820 g and Apgar scores between 6–8. All newborns were transferred to a third-degree Neonatology ICU service due to their prematurity. The fourth newborn was not identified in any of the ultrasounds performed during pregnancy. During the postpartum period, the patient had a fulminant evolution of COVID-19 pneumonia, with rapid deterioration, needing respiratory support and antiviral treatment. *Discussions*: Managing high-risk obstetrical pregnancies associated with COVID-19 requires a multidisciplinary team consisting of obstetricians, anesthesiologists, neonatologists, and infectious disease doctors. *Conclusion*: Our case is the first to our knowledge in Romania to present an association of high-grade multiple pregancy with COVID19 moderate form, rapidly evolving postpartum, needing rapid intensive care admission, and specific treatment with Remdesivir, with good post-treatment evolution.

## 1. Introduction

Higher-order multiple pregnancies represent a category of very high-risk pregnancies, and their number has increased in recent decades due to the greater use of assisted reproductive techniques (ART) [1]. Results of a study performed in 1997 considering higher-order multiple pregnancies showed that 43% were due to assisted reproductive technology, 38% were due to ovulation-inducing drugs, and 20% were conceived naturally [2]. Obviously, the risks are higher for the mother as well as for the fetuses. The impact on perinatal mortality was mostly due to prematurity and low birth weight. Chorionicity is crucial in higher-order multiple pregnancies, and monochorionicity, in particular, is associated with more life-threatening complications for the fetuses, such as feto-fetal transfusion syndrome, intrauterine death of one or more fetuses, and higher morbidity and mortality rates due to prematurity at birth [3]. Quadruplet pregnancies included in this category are associated with multiple potential complications among which the most frequent one is preterm delivery with a median gestational age at birth between 28 and 31 weeks. High multiple pregnancies are associated with an increased number of complications and most of the time represent a challenge for the obstetrical team [4,5].

Such a situation for a patient with other pathological antecedents might be considered even more of a challenge during the COVID-19 pandemic period. Since the beginning of 2019, more than 100 million individuals have been infected and approximately 2.4 million have lost their lives due to severe acute respiratory syndrome coronavirus 2 (SARS-CoV-2) [6]. When compared with nonpregnant patients, pregnant patients are more prone to be admitted into intensive care units and to go through invasive ventilation with a higher mortality rate. On the other hand, when comparing the nonpregnant and pregnant populations it seems that diabetes and obesity are risk factors in both groups [7]. During the third trimester of pregnancy, physiological changes in pulmonary function occur. The functional residual capacity is decreased by approximately 20–30% and oxygen consumption is increased; therefore, respiratory infections are not well tolerated. COVID-19 symptomatology for pregnant patients is similar to that of the general population; after the incubation period, primarily respiratory, gastrointestinal, and neurological symptoms start to appear with great variability in severity. Compared with nonpregnant patients, approximately 5% of the infected pregnant patients will develop a severe to critical illness [8]. The main issue with COVID-19 is that compared with other viral infections like influenza, clinical decompensation will happen relatively late in the course of illness [9].

## 2. Case Report

We report the case of a 37-year-old pregnant primipara who was admitted to the obstetrics service of “Elena Doamna” Hospital of Obstetrics and Gynecology–Iasi with the diagnosis of 34 weeks of gestation, triplets, threatened premature birth, SARS-CoV-2 infection, paroxysmal supraventricular tachycardia, and atrial septal defect surgically treated during childhood.

The pregnancy was obtained by an assisted reproductive technique (intrauterine insemination with ovulation induction) performed at another facility. The maternal cardiac pathology was considered minor and safe for the usage of assisted reproductive technology. No embryonic reduction was performed due to the patient’s choice. For the moment state law allows patients to choose whether they want or not to opt for embryonic reduction. The patient was previously admitted to another obstetrical hospital at 27 weeks of gestation for threatened premature birth where she was kept under tocolytic treatment, and she received antenatal corticosteroid therapy for fetal maturation. A day before admission to our service, the patient started developing a mild fever of 38 °C, dry cough, fatigue, and rigors. A rapid antigen test for COVID-19 was performed showing a positive result, and then COVID-19 polymerase chain reaction testing confirmed the infection. On admission to our unit, the patient was at 33 weeks and 3 days of gestation, hemodynamically stable (blood pressure 100/80 mm·Hg, heart rate 85 beats per minute, oxygen saturation of 98%) with mild hypogastric pain and mild dry cough and rigors. The vaginal examination showed the presence of an Arabin pessary and a uterine cervix with closed external orifice, leucorrhea of physiological aspect and in normal quantity, and at the vaginal digital examination the uterine cervix was shortened, with a soft consistency and a closed external orifice. At the fetal ultrasound performed by the hospital’s ultrasound specialist, three fetuses were observed with weights between 1400–1800 g, normal amniotic fluid indexes, three placentas, and three amniotic sacs; all fetuses had fetal cardiac frequency in the normal range and were in good general condition. The cardiotocography done at admission showed low intensity and irregular contractions. The patient was placed under tocolytic therapy with antispastics and progesterone administered locally. She continued her cardiac treatment with 150 mg Propafenone 2 tablets per day and 75 mg aspirin per day.

Thoracic radiography was performed at admission showing no signs of pneumonia. On the second day of hospitalization, the patient started having regular and high-intensity systematic uterine contractions that did not respond to any tocolytic treatment. After discussing the case with a multidisciplinary team composed of obstetricians, neonatologists, and anesthesiologists, the decision was made to perform a cesarean operation. Considering the risk of maternal respiratory decompensation, unresponsiveness to tocolytic treatment, and high order multiple pregnancy the multidisciplinary team opted for a preterm caesarian instead of risking maternal or fetal life. The Arabin pessary was removed and then the cesarean intervention was performed. The first baby was extracted in a complete pelvic presentation, masculine sex, weighing 1400 g and receiving an Apgar score of 7 and 8 at 5 min. The second baby was extracted in cephalic presentation, masculine sex, weighting 1600 g and receiving an Apgar score of 8. The third baby was extracted in cephalic presentation, masculine sex, weighting 1820 g and receiving an Apgar score of 7 and 8 at 5 min. The fourth baby was extracted in transverse presentation through internal version, masculine sex, weighting 1520 g and receiving an Apgar score of 6 and 8 at 5 min. The presence of a fourth fetus was not indicated during previous morphological ultrasounds, but this represents one of the limitations of ultrasound evaluation in higher-order multiple pregnancies. All four newborns tested negative for COVID-19 infection, tests were done with Real-Time PCR. During the cesarean intervention, no complications were encountered, and the hemostasis and uterine retraction were effective.

After delivery, all four babies were immediately transferred to a third-degree neonatology service due to their prematurity and need for special medical care. The post-intervention obstetrical evolution was excellent, the patient was stable (blood pressure 120/80 mm·Hg, heart rate of 70 bpm, oxygen saturation 98%) with a hemoglobin of 10.54 g/dL, normal leukocytes, minor thrombocytopenia (137,000/mm^2^), tonic uterus, lochia of a normal aspect according to the immediate postpartum period, physiological intestinal transit, and diuresis and normal body temperature. Antibiotic treatment with ceftriaxone 1 g at 12 h was initiated the day of the cesarean delivery and continued until the day of transfer to the infectious disease service. A low molecular weight heparin 5000 UI/day was administered as well throughout the patient’s hospitalization along with pain relief medications, acetaminophen, and nausea relief drugs.

Third day postpartum, the patient was moderately dyspneic, and dry coughing increased with a significant decrease in oxygen saturation (92%). Oxygen therapy with 2–3 L/min flow via nasal cannula was established, and dexamethasone 6 mg at 8 h with gastric protection (omeprazole 20 mg/day) was initiated. The thoracic radiography (Figure 1A) shows confluent opacities that were extended at three-quarters of the pulmonary area with aspects of bilateral COVID-19 pneumonia. No liquid was observed at the pleural level. Fourth day postpartum, the patient became unstable, with a further oxygen saturation decrease (86%) under 2–3 L at mobilization; therefore, the oxygen dosage was increased to 8–10 L, which increased the oxygen saturation to 96%. In collaboration with the infectious disease service, the decision to start Remdesivir therapy was made. Respiratory symptomatology increased and was refractory to the treatment. Oxygen delivery was changed to a nonrebreather mask. Laboratory exams showed elevated liver enzymes ALT/TGP 47 U/L and AST/TGP 51/L, glycemia 171.6, calcium 7.96 mg/dL, alkaline phosphatase 244.1, and lactate dehydrogenase 533 µg/dL. The patient was transferred to the intensive care unit of the Infectious Disease Service–COVID-19 department where she required very high oxygen flow output and continuation of antiviral therapy. After 7 days, the patient was stable with good evolution, and she was discharged 14 days later, with no post-treatment complications.

## 3. Discussion

It has been proven that high-order multiple pregnancies are associated with an adverse impact on perinatal morbidity and mortality [10]. Attempts have been made to create strategies to prevent such pregnancies. Strategies like canceling cycles of ovulation induction when there are more than two mature follicles, but these would, unfortunately, reduce the pregnancy rate per cycle and imply even higher economical costs [11,12].

A review study conducted in 2009 that included 130 articles concerning high-order multiple pregnancies identified certain risk factors for obtaining these types of pregnancies. The risk factors described were seven or more preovulatory follicles with dimensions higher than 10–12 mm, estrogen values higher than 1000 pg/mL, early cycles of treatment, patients younger than 32 years old, a low body mass index, and the use of donor sperm. Nonetheless, at the end of this study, the author concludes that the cancellation of any cycle treatment would be counterproductive [13]. Our patient obtained the pregnancy at 37 years of age and had a normal body mass index. As challenging as it is to implement obstetrical management for high-order multiple pregnancies, it is even more challenging to do this for a patient suffering from COVID-19 infection. For asymptomatic patients infected with COVID-19, there is usually no need for any other particular laboratory investigations when compared with a normal pregnant patient. On the other hand, for symptomatic patients with significant respiratory symptoms or systemic illness, a thorough investigation should be performed, and other particular laboratory investigations were taken into consideration, such as C-reactive protein (CRP). Confusingly, CRP normally rises during pregnancy, making it more difficult to create a clinical trajectory by only considering this parameter [14]. Ferritin testing was indicated by The American Society for Maternal-Fetal Medicine in patients with body temperatures higher than 39 °C undergoing antipyretic treatment because it may predict a cytokine storm syndrome [15]. Measuring the D-dimer levels is not indicated for patients who do not need hospitalization and for pregnant patients because their values may vary due to gestational status. Their interpretation may be misled during pregnancy [16]. Investigating D-Dimer and Interleukin-6 levels combined in pregnant patients with COVID-19 infection has suggested that elevated levels were associated with more severe forms and found to be present in 80% of critically ill patients and 60% severely ill patients [17]. Unfortunately, information considering levels of these laboratory tests during pregnancy are limited and comparisons are difficult to assess [18,19].

Administration of antibiotics or antiviral treatment is still being researched and improvements are being made every day. From the beginning of the COVID-19 pandemic in 2020, multiple therapeutic schemes have been displayed, implemented, and then withdrawn. Most of them targeted the general population and few the pregnant population. All therapeutic schemes indicated for pregnant patients were approved under the US Food and Drug Administration’s Emergency Use Authorization in consultation with The Society for Maternal-Fetal Medicine. Lately, it has been indicated to administer Remdesivir, an intravenous nucleotide prodrug of an adenosine analog for pregnant or postpartum patients with oxygen saturation lower than 94% on ambient air or those who require supplemental oxygen [15]. Anticoagulation therapy is mostly indicated in pregnant patients or the postpartum period for severe or critical cases of COVID-19 with a high risk for venous thromboembolism. The most used anticoagulants are the low molecular weight heparin or unfractionated heparin [20,21]. Our patient was under anticoagulation therapy with low molecular weight heparin due to the post-surgery status and rapid evolution of pneumonia and was also receiving antibiotics, antipyretic, pain drugs, and medication for nausea.

A recent study performed by Ellington S. et al. identified an increased risk for severe illness and higher chances for mechanical ventilation in pregnant patients suffering from COVID-19 infection when compared to other pregnant patients of the same age, race, and diagnosed with the same type of comorbidities [22]. More than 90% of pregnant patients that require Intensive Care Unit (ICU) admission due to COVID-19 infection are during their third trimester [17].

The fact that our patient had a cardiac pathology even though kept well under control made this case even more difficult to manage. The risk of developing cardiomyopathy due to COVID-19 infection is approximately 7–33% in the general population, but information regarding pregnant patients is limited [18,19]. Only a small case series reported the development of cardiomyopathy in two infected pregnant patients [23]. Due to the novelty of this viral infection and insufficient data regarding pregnant patients, it is difficult to suggest that COVID-19 related cardiomyopathy is increased in pregnant patients when compared to the general population.

One of the largest cohort studies performed studied the courses of 242 COVID-19 positive pregnant women as well as their 248 infants during the third trimester of pregnancy and the first month postpartum. The main outcomes of this study showed a higher rate of cesarean delivery and premature birth in patients hospitalized due to symptomatic forms of infection. Preterm deliveries occurred in one-third of pregnant patients with COVID-19. Out of these, 60% were late preterm deliveries (between 34 weeks and 36 weeks and 6 days of gestation) and 40% were early preterm deliveries (between 24 weeks and 33 weeks and 6 days of gestation) [17].

Data regarding fetal complications due to COVID -19 infection is limited, according to several studies miscarriage is more common during the first trimester (16.1%) when compared to the second trimester of pregnancy (3.5%) [24,25,26,27]. It is difficult to offer detailed data concerning fetal abnormalities secondary to maternal infection with COVID-19 but some studies have speculated the possibility of abnormal fetal growth due to placental insufficiency. Placental insufficiency might be a consequence of intervillous inflammation and thrombosis of fetal intervillous vessels as well as due to uteroplacental vascular malperfusion [28].

High multiple-order pregnancies are more often encountered nowadays due to the frequent use of artificial reproductive techniques. Their obstetrical management is difficult and requires special maternal-fetal surveillance. The disadvantages of the modern world’s rapid evolution imply also the rapid transmission of infectious diseases such as COVID-19 that led to a pandemic situation. Dealing with pregnant patients positive for this new viral infection has been a challenge to all medical fields and especially the obstetrical field. It is highly difficult to establish an antiviral treatment when it comes to pregnancy; all medications must be carefully chosen to not affect the fetus. Studies to see if a vertical transmission of this infection is passed on to the newborn and its effects on the infant are still needed and are impatiently awaited.

## 4. Conclusions

Managing cases of high-risk obstetrical pregnancies associated with COVID-19 needs a multidisciplinary team consisting of obstetricians, anesthesiologists, neonatologists, and especially an infectious disease physician.

Our case is the first to our knowledge to present such an association of high-grade multiple pregnancy with COVID-19 in a moderate form rapidly evolving postpartum, but with good response to specific antiviral treatment.

## Figures and Tables

**Figure 1 medicina-57-01186-f001:**
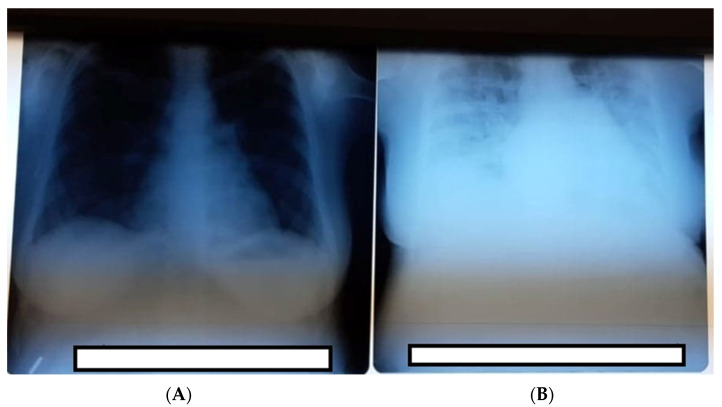
(**A**) Thoracic radiography was performed at admission with no obvious lesions. (**B**) Thoracic radiography third-day postpartum showing signs of severe COVID-19 pneumonia.

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
