# Peer review of "The Rare Case of a COVID-19 Pregnant Patient with Quadruplets and Postpartum Severe Pneumonia. Case Report and Review of the Literature"

_medicina, 2021, doi:10.3390/medicina57111186_

Round 1

Reviewer 1 Report

It is a well written case report on incidence of COVID-19 in a patient pregnant with quadruplets during last trimester of pregnancy.

Medical procedures undertaken to protect both the mother and babies are elaborated.

It would have been good to include the COVID-19 status of the babies as there have been reports that the incidence of COVID-19 infection in babies delivered by Caesarean section is slightly elevated.

Author Response

Dear Sir/Madam,

I would like to thank you for the comments and suggestions offered, they are indeed helpful in producing articles of higher quality. In response to your review we would like to inform you that all four new borns tested negative for COVID-19 infection. All testings were done using Real-Time PCR. 

Kind regards,

Akad Mona. 

Reviewer 2 Report

Interesting case report for Covid pandemic. 

In the "Discussion" part, it is better to focus on the effects of Covid pandemic on pregnant patients and to analyze other similar case reports. It is a case report on Covid, not on multiple pregnancy.

Author Response

Dear Sir/Madam,

I appreciate the suggestions and tried to modify the Discussion section in order to focus more on COVID-19 effects on pregnant patients as well as finding case series. 

Kind regards,

Akad Mona

Reviewer 3 Report

The Authors report an interesting case of quadruplet gestation and its management after COVID-19 symptomatic diagnosis. Although , the case is well presented , there are some punctuation and English corrections to do . Line 51: I suggest obstetrical team instead of obstetrical team in charge. Three questions : -maternal cardiac pathology is a contre-indication to high-risk pregnancy and to assisted reproductive technology ? Why no embryonic reduction was performed - I recommend to the Authors to precise if the cesarean section was recommended due to quadruplets pregnancy and why they did not perform CS after Sars Cov 2 negativation? - Infants were COVID-19 positives or negatives? Although , the case is well presented , there are some punctuation and English corrections to do . Line 51: I suggest obstetrical team instead of obstetrical team in charge. Three questions : -maternal cardiac pathology is a contre-indication to high-risk pregnancy and to assisted reproductive technology ? Why no embryonic reduction was performed - I recommend to the Authors to precise if the cesarean section was recommended due to quadruplets pregnancy and why they did not perform CS after Sars Cov 2 negativation? - Infants were COVID-19 positives or negatives?

Author Response

Dear Sir/Madam

Thank you for the comments and suggestions made in order to produce a higher quality article.

  1. Line 51 -obstetrical team in charge was changed to obstetrical team.
  2. The maternal cardiac pathology was considered as a minor risk for assisted reproductive techonology,  it did not represent a contre-indication, the patient was hemodinamically stable until COVID-19 infection.
  3. No embryonic reduction was performed as a patient choice. State law allows patients to chose weather their want or not to opt for embryonic reduction.
  4. The casearean operation was perfomed  due to the posibility of rapid maternal respiratory decompensation in combination with high order multiple pregnancy and non responsiveness to tocolytic therapy.  Waiting until Sars Cov 2 negativation implied risks for maternal and fetal lifes. 

All modifications required were inserted in the article text.

Kind regards,

Akad Mona. 
